# Volcanic ash leaching alters the trace metal distribution within the coral holobiont of *Stylophora pistillata*

Frank Förster<sup>1\*</sup>, Sebastian Flöter<sup>1</sup>, Lucie Sauzéat<sup>2,3</sup>, Stéphanie Reynaud<sup>4</sup>, Eric Achterberg<sup>5</sup>, Alexandra Tsay<sup>1</sup>, Christine Ferrier-Pagès<sup>4</sup>, Tom E. Sheldrake<sup>1</sup>

- 5 <sup>1</sup>Department of Earth Sciences, University of Geneva, Genève, Switzerland
  - <sup>2</sup>Laboratoire Magmas et Volcans (LMV), Université Clermont Auvergne, CNRS, IRD, OPGC, F-63000 Clermont-Ferrand, France
  - <sup>3</sup>Institut de Génétique, Reproduction et Développement (iGReD), Université Clermont Auvergne, CNRS, INSERM, F-63000 Clermont-Ferrand, France
- O <sup>4</sup>Ecophysiology Team, Centre Scientifique de Monaco, Monaco, Monaco
  - <sup>5</sup>GEOMAR Helmholtz Center for Ocean Research Kiel, Kiel, Germany

Correspondence to: Frank Förster (Frank.Foerster@unige.ch)

Abstract. Explosive volcanic eruptions generate large amounts of volcanic ash that release essential and nonessential trace metals upon deposition in seawater, modifying its chemical composition. Tropical scleractinian corals, known for accumulating trace metals, are susceptible to these changes, making them valuable biomonitors for increased metal concentrations. In this study, we investigated how volcanic ash leaching influences trace metal partitioning within the hermatypic branching coral Stylophora pistillata through six-week coral culture experiments. Coral nubbins were reared under control and ash exposed conditions, with 2.5 g ash added three times a week (averaging 250 mg L<sup>-1</sup> per week). We quantified trace metals (V, Mn, Fe, Co, Ni, Cu, Zn, Cd, and Pb) in the ash-seawater leachate, and in three distinct coral compartments (skeleton, tissue and symbionts). 24 hour ash leaching experiments at a ratio of 1:100 (g ash: mL seawater) demonstrated that ash from La Soufrière (St. Vincent) released trace metals in the order Mn, Zn, Co, Cu, Cd, Fe, and Ni into seawater, while Pb and V were scavenged. Trace metal concentrations in coral compartments correlated with seawater concentrations, with most significant changes observed in the skeletal metal content. Ash exposure enriched skeletal concentrations of V, Mn, Fe, Ni, and Zn while depleting Cu and Pb. Ash leaching also shifted the metal distribution in coral skeletons, affecting relationships between transition and alkaline earth/alkali metals. Apparent skeletal distribution coefficients  $(K_{El})$  for labgrown corals showed most trace metals were less abundant in skeletons than seawater  $(K_{El} < 1)$ , except for Pb, Cd and Co ( $K_{El} > 1$ ). Metal concentrations varied between tissues and symbionts, with Mn and Fe significantly enriched in ash exposed tissues. Volcanic ash releases a range of trace metals, altering the coral metallome by affecting bioaccumulation and metal redistribution across coral compartments. These findings not only advance our understanding of coral trace metal dynamics at the organismal level, but also provide a basis for estimating environmental metal fluxes during future eruptive scenarios and help interpret geochemical coral archives of past eruptions.

# 1 Introduction

In marine environments, trace metals such as vanadium (V), manganese (Mn), iron (Fe), cobalt (Co), nickel (Ni), copper (Cu), zinc (Zn), cadmium (Cd), and lead (Pb) have a significant biological importance. At the organismal level, they serve as essential cofactors in various metalloenzymes (Raven et al., 1999; Andresen et al., 2018), but can also cause toxicity, harming marine organisms even at low concentrations (Dal Pizzol et al., 2022; Chakraborty et al., 2010). Certain trace metals can bioaccumulate along the aquatic food chain (Bosch et al., 2016; Sonone et al., 2020), with their uptake and toxicity governed by factors such as metal availability (Reich et al., 2020), interactions between trace metals (Morel and Price, 2003; Rodriguez et al., 2016) and chemical speciation (Allen et al., 1980). This underscores the importance of understanding their dynamic behaviour in marine ecosystems.

Every year, hundreds of millions of tons of inorganic airborne particulates such as desert dust are transported globally and subsequentially deposited on land and in the ocean (Garrison et al., 2003). These dust events can supply essential and nonessential trace metals to open ocean environments, consistently modifying the chemical composition of the seawater (Torfstein et al., 2017). While far less frequent and irregular, volcanic eruptions can induce rapid environmental changes that range from local to global scales. Depending on the eruption size, explosive volcanic eruptions are capable of ejecting  $\approx 10^{11}$  –  $10^{15}$  kg of tephra (fragmented magma in the form of rocks and fine particles) into the atmosphere (Self, 2006). Volcanic ash is the finest tephra, with particle diameters <2 mm, and is produced during explosive volcanic eruptions. It can be transported across hundreds to thousands of km before settling (Pyle et al., 2006). Tephra deposition is a known source of various essential and nonessential metals in ocean surface waters (Olgun et al., 2011), which was demonstrated experimentally through ash leaching experiments (Hoffmann et al., 2012; Frogner et al., 2001; Wygel et al., 2019) and in field sampling campaigns (Achterberg et al., 2021; Censi et al., 2010).

Ash leaching studies are commonly performed to extract water soluble compounds adsorbed onto ash surfaces, to quantify the potential hazard of volcanic eruptions, for example the drinking water supply (Stewart et al., 2006). Flow through and batch leaching experiments with various different ashes, reveal that volcanic ash generally releases a wide range of major and minor elements into the leaching solution (Stewart et al., 2020; Wygel et al., 2019; Ayris and Delmelle, 2012). The release of metals from volcanic ash occurs through two main mechanisms: the dissolution of water-soluble metal salt coatings on the surface of ash particles (Óskarsson, 1980; Jones and Gislason, 2008; Duggen et al., 2010) and the partial dissolution of the silicate glass and mineral components of the ash (Gislason and Hans, 1987). Salts are extremely thin sulphate and halide deposits (< 10 nm) that precipitate at the ash-liquid interface, when freshly erupted ash interacts with acidic gases (i.e. SO<sub>2</sub>, HCl, HF) and aerosols (i.e. H<sub>2</sub>SO<sub>4</sub>) in the plume (Delmelle et al., 2007). For Fe, Olgun et al. (2011) found no positive correlation between the trace metal content of ashes and the amount of the trace metal released into the seawater, with ashes containing lower Fe (<6 wt.%) generally releasing more soluble Fe than those with higher Fe content (>8 wt.%). This suggests that the rapid release of Fe is not primarily driven by the relatively slow dissolution of volcanic glass and/or minerals in seawater, but rather by fast surface reactions, such as the dissolution of metal salt coatings (Duggen et al.,

2010)The most significant leaching typically occurs within the first ten minutes of contact with the leaching solution, as the surface metal salts rapidly dissolve in water (Frogner et al., 2001; Tomasek et al., 2021), while steady state dissolution for various volcanic ashes was achieved by 24 h leaching (Tomasek et al., 2021). In addition, to release trace metals into seawater, airborne volcanic ash can preferentially scavenge certain metals (Staudigel et al., 1998; Rogan et al., 2016). Scavenging occurs when ions are selectively adsorbed onto surfaces of ash and other particles and continues until the particles become saturated with surface associated ions. High silica volcanic glass, a major constituent of many volcanic ashes, has been shown to preferentially adsorb higher valence ions onto its surface, presumably through cation exchange reactions (Steinhauser and Bichler, 2008).

Tropical coral reefs are often exposed to metals deposited by dust clouds (Shinn et al., 2000; Blanckaert et al., 2022), or ash fall (as projected for the coral reefs in the coral triangle (Fischer, 2023)). Although volcanic ash is a potent source of various essential trace metals (Förster et al., 2024), the ash can have detrimental effects on aquatic ecosystems (Hoffmann et al., 2012). Stewart et al. (2020) argue that ash deposition probably does not cause metal toxicity. Instead, they suggest that the major impacts of suspended ash are physical, including habitat smothering, increased turbidity (e.g. Lallement et al., 2016; Witt et al., 2017) and ocean acidification (Wall-Palmer et al., 2011). Scleractinian corals, which are the main reef builders, are susceptible to changes in seawater trace metal compositions and are bioindicators of water quality (Ranjbar Jafarabadi et al., 2018). As they need several metals for growth and other metabolic processes, they accumulate trace metals from the environment (Esslemont et al., 2000; Runnalls and Coleman, 2003; Esslemont et al., 2004), offering valuable insights into shifts in seawater chemistry. Subtle changes in trace metal concentrations can directly impact symbiont photosynthesis and growth rates (Biscere et al., 2018; Ferrier-Pagès et al., 2005; Ferrier-Pagès et al., 2001) and influence their resilience to abiotic stressors (Amorim et al., 2024). The coral holobiont, a complex meta organism consisting of a chidarian animal, dinoflagellate endosymbionts from the Symbiodiniaceae family (Lajeunesse et al., 2018) and other microbial partners (Matthews et al., 2020), can accumulate trace metals in each of its components. The photosynthetic dinoflagellates have a special role in trace metal accumulation and metal exchange within the organism (Reich et al., 2023), as their uptake can surpass that of coral tissue by an order of magnitude (Blanckaert et al., 2022; Ranjbar Jafarabadi et al., 2018; Reichelt-Brushett and Mcorist, 2003; Ferrier-Pages et al., 2018). Metal concentrations in coral soft tissues are dynamic, with rapid bioaccumulation occurring within the first days of contamination, followed by a slower, metal specific recovery in noncontaminating conditions, which can take over 40 days (Metian et al., 2015). This is influenced by the sequestration of metals into the mucus (Reichelt and Jones, 1994) and skeleton (Metian et al., 2015). However, the elemental partitioning from tissue to coral skeleton is metal specific but still poorly understood (Esslemont et al., 2000). Understanding the accumulation of metals in the coral skeleton is of major importance as it is extensively used as an archive of past environmental conditions, based on the incorporation of different trace elements (e.g. Saha et al., 2016; Jiang et al., 2022; Anu et al., 2007b; Esslemont et al., 2004). Various geochemical tracers (isotopes and element/Ca ratios) in coral skeletons have been utilized to reconstruct ambient seawater properties, such as temperature (Hathorne et al., 2013; Montagna et al., 2014; Corrège, 2006; Tierney et al., 2015), pH (Hönisch et al., 2004; Mcculloch et al., 2012) and sedimentary input (Yu et

al., 2022).

We conducted controlled coral culture studies to better understand how a volcanic ash deposition event may impact the physiology and metal uptake of reef building corals. While the physiological coral response to ash exposure was previously published, which included analysis of tissue metal and stable isotope contents (Förster et al., 2024; Förster et al., 2025), the present study expands these findings by focusing on the intra-organismal (skeleton, host tissue, and symbionts) trace metal distribution (V, Mn, Fe, Co, Ni, Cu, Zn, Cd, and Pb) of the branching coral *Stylophora pistillata*. To achieve this, ash-seawater leaching experiments were conducted to quantify the geochemical input of pristine ash into seawater, a compartment model was developed to simulate real time metal concentrations in the culture tanks, and the trace metal content of coral skeletons originating from the culture experiments were analysed. This multi-faceted study provides critical insights into trace metal cycling within the coral holobiont after post volcanic changes in seawater metal concentrations. It contributes to the application of scleractinian corals as reliable biomonitors and natural archives of water quality and environmental change, including volcanic eruptions.

### 2 Materials and Methods

# 2.1 Experimental setup

### 2.1.1 Ash leaching experiments

Pristine freshly erupted volcanic ash (i.e., not in contact with water) was collected from a balcony in central Barbados on the 9th of April 2021 by Dr. John B. Mwansa, adhering to the collection and storage suggestions outlined by Witham et al. (2005) and Stewart et al. (2020). The ash originated from the 2021 explosive eruption of the La Soufrière volcano on St. Vincent, resulting in an ash plume reaching heights of 8-15 km (Joseph et al., 2022). Significant andesitic-basaltic ash deposits ranging in thickness from 1-10 mm, comprising predominantly of volcanic glass and plagioclase (Horwell et al., 2023), were observed on Barbados, approximately 175 km east of the eruption site. Barbados and St. Vincent are situated in the Eastern Caribbean and host actively growing coral reefs (Lewis, 1960; Adams, 1968), all of which were exposed to volcanic ash following the 2021 eruption. To assess seawater metal concentrations after ash exposure, leaching experiments were conducted on the unsieved volcanic ash, with a median particle size of 40.6 μm and a sorting of 43.65 μm, calculated according to Inman (1952). The experimental setup follows the standardized protocol for ash hazard characterization by Witham et al. (2005) and Stewart et al. (2020). The extraction method and equipment used are presented in Table 1. To achieve a ratio of 1:100 (1g volcanic ash: 100 mL seawater leaching solution), 30 mL of freshly collected 0.2 µm filtered seawater was used as leaching solution and mixed with 0.3 g of pristine volcanic ash in a 50 mL centrifuge tube. Procedural seawater samples were taken as blanks. The 24 h ash leachate and seawater blanks were taken in triplicates, whilst the 1 h and 30 d ash leachates were sampled once. After centrifugation and filtration of the leachate, the solution was acidified to 0.5 M HNO<sub>3</sub> (NORMATOM® from VWR) and stored at 4°C until analysis.

Table 1: Experimental setup and conditions for ash leaching

| Extraction vessel   | acid-cleaned 50 mL                 |  |  |
|---------------------|------------------------------------|--|--|
|                     | polypropylene (PP)                 |  |  |
|                     | centrifuge tubes                   |  |  |
| Agitation<br>method | Digital platform rocker            |  |  |
|                     | (Thermo Scientific <sup>TM</sup> ) |  |  |
|                     | rocking at $20^{\circ}$ and 45 rpm |  |  |
| Leaching            | 1 h, 24 h, 30 d                    |  |  |
| duration            |                                    |  |  |
| Centrifugation      | 5500 rpm                           |  |  |
| speed               | 5500 Ipili                         |  |  |
| Centrifugation      | 10 min                             |  |  |
| time                | 10 111111                          |  |  |
| Filtration          | Syringa filtration                 |  |  |
| method              | Syringe filtration                 |  |  |
| Filters used        | $0.22~\mu m$ polyethersulfone      |  |  |
|                     | (PES)                              |  |  |

The ash leachate solution was analyzed using a high-resolution inductively-coupled plasma mass spectrometry (HR-ICPMS) (Element  $XR^{TM}$ , Thermo Scientific) at GEOMAR, Helmholtz Centre for Ocean Research Kiel. Samples were diluted 12.5 times with deionized water, before preconcentration of trace metals and the removal of the seawater matrix was performed using SeaFAST (Rapp et al., 2017). The experimental procedure follows Rapp et al. (2017), with the exception that a NOBIAS Chelate-PA1 resin column (200  $\mu$ L) was used for preconcentration. Trace metal concentrations of V, Fe, Ni, Cu, Zn, Cd and Pb were quantified using isotope dilution and standard addition for Co and Mn. Accuracy was assessed on certified seawater reference materials (CASS-6 and SAFe D1\_479) with recoveries ranging from 87% for Fe to 110% for V, with an average deviation < 5% from consensus values (Tab. S1). Reproducibility was assessed on triplicate analyses of samples with most RSD < 15%, except for Fe and Co with a RSD of 47% and 29%, respectively. Trace metal concentrations are presented as [nmol L-1] or [mg L-1].

# 2.1.2 Coral culture experiments

A detailed description of the coral culture conditions and the experimental setup is given in Förster et al. (2024). In brief, 42 coral fragments ("nubbins") were cut from large parent colonies of the scleractinian branching coral *S. pistillata*, which were consistently reared in culture conditions (light:  $200 \pm 10 \mu mol$  photons m<sup>-2</sup> s<sup>-1</sup>; temperature:  $25 \pm 0.2$  °C; pH<sub>SWS</sub>:  $7.93 \pm 0.04$ , salinity: 38.5 %) at the Centre Scientifique de Monaco (Monaco). This species was chosen due to its widespread geographical distribution, its well-known physiology and responses to various stressors, including metals (Ferrier-Pagès et al., 2003; Ferrier-Pagès et al., 2005; Hoogenboom et al., 2012; Biscere et al., 2015; Metian et al., 2015; Biscere et al., 2017; Biscere et al., 2018).

Corals were fed with *Artemia salina* nauplii during the healing phase. Feeding was stopped one week before and during the experiment to avoid possible interactions between naturally containing metals in the food source and ash derived metals. For the experiment, the nubbins, each approximately 2-4 cm in size, were evenly distributed among four tanks (30 L) and maintained under two different conditions, with two tanks per condition: a control condition and an experimental condition in which they were exposed to volcanic ash for six weeks. The experimental runtime was similar to the median eruption time of seven weeks (Siebert et al., 2015). Three times a week, 2.5 g of unfiltered volcanic ash was added as suspension into two of the experimental tanks (averaging 250 mg L<sup>-1</sup> per week). As a preleaching step, the ash was mixed in a 50 mL centrifuge tube filled with 0.2 µm filtered seawater for 24 h before the addition. The ash enrichment was selected to have little effect on shading and did not cause polyp smothering (Förster et al., 2024), as the focus of this study was to understand how healthy corals incorporate metals supplied by volcanic ash. Each morning, the seawater inlet to the tanks was closed, and the volcanic ash suspension was added three times a week to the experimental tanks. After eight hours, the water supply was resumed, providing a complete water renewal overnight. Tank maintenance comprised the removal of ash deposits and algae, which occurred weekly to prevent coral overgrowth by filamentous algae. After the six weeks of experimental runtime, coral nubbins were stored at -20 °C for further analysis.

### 2.2. Trace metal determination in culture tank seawater and coral soft tissue

### 2.2.1 Modelled trace metal concentrations in culture tanks

For the purpose of defining the real time trace metal concentrations in the ash exposed seawater within a coral culture tank (30 L), we developed an empirical compartment model utilizing the deSolve package (Soetaert et al., 2010) in RStudio Version 4.2.0 (Team, 2022). This model presents the time dependent individual metal concentrations in the tank (1):

$$\frac{dC_{tank}}{dt} = \left(\frac{Q_{in}}{V} \cdot \frac{dC_{in}}{dt}\right) - \left(\frac{Q_{out}}{V} \cdot \frac{dC_{out}}{dt}\right) + \frac{\varphi(t)}{V} \tag{1}$$

in which the flux of seawater, with the blank seawater metal concentration ( $C_{in}$  and  $C_{out}$ ), into ( $Q_{in}$ ) and out of ( $Q_{out}$ ) the tank volume (V) is identical, following a stepwise profile between 0 and 200 mL/min (Fig. 1).  $\varphi(t)$  represents the time dependent flux of metals leached from volcanic ash at minutely time steps (t). We described the long term leaching of

volcanic ash by a single mode first order decay model (2) (Kim et al., 2018; Yin et al., 2020), which assumes a constant desorption of exchangeable metals into the leaching solution over time (*t*).

$$\varphi(t) = A_0 e^{-kt},\tag{2}$$

where  $A_0$  is the maximum leachable quantity from ash [nmol L<sup>-1</sup>] and k is the ash dissolution constant [nmol L<sup>-1</sup>per min], which is approximated by fitting the curve using the results of the ash leaching experiments after 0 h, 1 h, 24 h and 30 d (Tab. S2). To obtain  $A_0$ , we needed to estimate the highest possible elemental concentration supplied by the ash in the coral culture tank, based on the results we obtained in the leaching experiments. For most elements,  $A_0$  is positive, indicating ash releases metals. For V and Pb, however,  $A_0$  is negative, suggesting scavenging (Tab. S2). Given that the ash metal discharge into solution is highest in the first few hours of the experiment (Jones and Gislason, 2008; Frogner et al., 2001; Wygel et al., 2019; Tomasek et al., 2021), and steady state dissolution is achieved by 24 h (Tomasek et al., 2021), a leaching time of 24 h was chosen as a good approximation of the total leachable fraction. To upscale the expected concentration from the leaching experiments (0.3 g ash : 30 mL seawater) to the coral culture experiments (2.5 g ash : 30 L seawater), we converted the trace metal concentrations after the 24 h leaching experiment into mass of elements leached per mass of supplied ash [mg kg<sup>-1</sup> ash] (3).

$$m_{El} \ leached/m_{ash} = \frac{M_{El} * c_1 * V_1}{m_1}$$
 (3)

with  $M_{El}$  the elemental molar mass [g/mol];  $c_l$  the difference in metal concentration [nmol L<sup>-1</sup>] between 24 h (0.3 g ash : 30 mL seawater) and the seawater blank;  $V_l$  the volume of the leachate solution in the falcon tube (30 mL); and  $m_l$  the mass of added ash (0.3 g). Given the relatively stable metal dissolution at low ash : leaching solution ratios (e.g. 1:100 to 1:1000; Tomasek et al., 2021), we assume that the 1:100 ratio in our leaching experiments is representative of the conditions in the aquarium (i.e. 1: 12000). This allowed us to calculate the moles of the elements supplied by the ash ( $n_{ash}$ ) (4):

$$n_{ash} = \frac{m_{El} \operatorname{leached}/m_{ash} * m_2}{M_{El}} \tag{4}$$

with  $m_2$  the amount of ash (2.5 g) added into the experimental tank ( $V_2 = 30$  L). The calculated concentration in a tank following 24 h of ash leaching ( $c_2$ ) is calculated as the sum of moles in the seawater blanks ( $n_{sw}$ ) and released by the ash ( $n_{ash}$ ), divided by  $V_2$  ( $c_2 = n_{ges}/V_2$ ). The maximum ash input ( $A_0$ ) can then be calculated with  $A_0 = c_2 - c_{sw}$ .

To account for the thrice weekly ash addition and tank cleaning schedule, the weekly flux of metals from volcanic ash depends on the time when the ash is added  $(t_{ashIn})$  and the time the tank is cleaned  $(t_{clean})$  (5). During tank cleaning  $(t_{clean})$  deposited ash is entirely removed from the tank, reducing the ash input to zero. Since tank cleaning was performed every Friday afternoon, just a few hours after the third weekly ash addition, the resulting peak in metal concentration was lower than that observed after the second ash addition (Fig. 1b).

$$ash\ input(t) = \sum_{t=1}^{168} \begin{cases} A_0 e^{-k(t-t_{ashIn})} & if\ t_{ashIn} \le t \le t_{tank\ clean} \\ 0 & if\ t > t_{tank\ clean} \end{cases} \tag{5}$$

Figure 1: Modelled flux of ash derived metals during coral culture experiments. a) Conceptual representation of the input parameters defining the system dynamics as showed in Eq. (1). Ash input is specified in Eq. (2) and exemplified over a one-week period. The thrice-weekly addition of ash results in cumulative effects from bulk ash leaching, which are entirely reset during the periodic tank cleaning. (b) Modelled Co concentration [nmol/L] in the coral culture tank throughout experimental period. The seawater concentration represents ambient Co levels in the Mediterranean Sea, while the mean indicates the averaged Co concentration over the entire runtime.

### 2.2.2 Metal analysis of coral host tissue and symbionts

The data presented in this section are already published in Förster et al. (2024). Briefly, three samples per tank (n = 6 per condition) were used to determine the metal concentration in coral host (tissue) and symbionts. Before the analysis, tissue (containing the symbionts) was separated from the skeleton using a mixture of compressed air with deionized water from an airbrush. The solution containing both tissue and symbionts was ground with an Ultra-Turrax T25 (Janke & Kunkel (IKA)®), and then centrifuged three times at 8000 G for 10 min and 4 °C. After each centrifugation step, the supernatant comprising the coral tissue was transferred to a separate 50 mL centrifuge tube and the concentrate containing symbionts resuspended with deionized water. Both symbionts and coral tissue were first freeze dried, and then dissolved in a concentrated HNO<sub>3</sub> (purity ≥ 65%, Puriss. p.a., Reag.ISO, Honeywell)-H<sub>2</sub>O<sub>2</sub> (purity ≥ 30%, Puriss. P.a., Perdrogen<sup>TM</sup>, Honeywell), mixture in Savillex beakers for major and trace element analyses. Dissolution was first conducted at room temperature for 24 h to avoid overpressure and loss by evaporation or leakage, and then at 100 °C for at least 72 h. To ensure complete digestion, this operation was repeated three times. After complete digestion, samples were first evaporated to dryness at 80 °C to avoid loss by volatilization of highly volatile elements like selenium (Se) (Vlastelic and Piro, 2022) and then, the residue was diluted in 10 mL 0.5 M HNO<sub>3</sub>. Weighed aliquots of this "mother" solution were diluted with 2% v/v HNO<sub>3</sub>. Note that major and trace element analyses were conducted at the Laboratoire Magmas et Volcans (LMV) in clean laminar flow hoods using double distilled acids to avoid exogeneous contamination. Major and trace element concentrations were measured by ICP-OES (Agilent 5800) and QQQ-ICP-MS (Agilent 8900), respectively, at the LMV, as described in Förster et al. (2024). Element concentrations were determined using calibration curves from blanks and a series of dilutions of mutli-element standard solutions. Accuracy and precision were evaluated through duplicate and re-run analyses of samples as well as the analysis of matrix-matched certified RMs (1577c, BHVO-1, BHVO-2, and DR-N), while Indium (In) served as internal standard to assess analytical drift. The measured values are within uncertainty of previously published results and reproducibility was generally better than 5%. Trace metal concentrations in coral tissue and symbionts are reported and presented as mass fraction [dry weight mg kg<sup>-1</sup>].

### 2.3 Trace metal analysis on coral skeletons

### 2.3.1 LA-ICP-MS measurements on coral apices

Three coral nubbins were selected from each tank for their skeletons (n = 6 per condition). From each skeleton, two coral apices (≈5mm length) were cut with a Dremel© electric hand drill and cleaned for 48 h in a 1:4 mix of household bleach solution (NaClO, 4%, Reactolab SA) and deionized water, following sample preparation by Tanzil et al. (2019). Apices were then sonicated in deionized water for a total of 30 min, while water was changed every 10 min. For drying, apices remained in an oven (40 °C) for 24 h. Dried samples were embedded in epoxy resin and ground with pristine Silicon Carbide paper (grit size P800 and P1200) until the cross section of the coral apex revealed a clear axial corallite (Fig. 2). 30 LA-ICP-MS spot analyses were performed per embedded coral tip, along multiple theca, totaling 750 point measurements on 24 branch tips (360 points on 12 control tips; 390 points on 12 ash exposed tips). The locations for point measurements were chosen within the outermost region of the coral apex, ranging from the tip's end to approximately 1-2 mm inside, ensuring that only recently formed skeleton (deposited during the experiment) was sampled. This selection was based on an assumed growth rate, averaging 0.04 mm per day for *S. pistillata* (Liberman et al., 1995; Shaish et al., 2006; Kotb, 2001), suggesting an approximate growth of 1.68 mm skeleton during the culturing period.

Fig. 2: S. pistillata apex skeleton. (a) Tomographic cross section of branching coral tip with highlighted architectural features: ac - axial corallite, cl - columella, ed - endo dissepiment, t - theca; (b) micrograph of an analyzed coral tip with 30 laser ablation pits following the theca of the axial corallite.

Measurements were performed with an ESL 193 HE laser ablation system coupled with an Agilent 8900 triple quadrupole mass spectrometer (LA-ICP-MS) at the Department of Earth Sciences, University of Geneva (Switzerland). Helium (He) was used as carrier gas at a flow rate of ≈850 mL/min. The instrument was tuned to ThO/Th < 0.3%, mass-21/mass-42  $\approx 0.2\%$  and  $^{238}\text{U}/^{232}\text{Th}$  of  $\approx 1.0$  using the glass reference material NIST610. The samples were ablated at an energy density of ≈3 J/cm<sup>2</sup>, repetition rate of 10 Hz, and a laser spot size of 60 µm. Isotopes were selected to avoid potential interferences in LA-ICP-MS analysis of carbonates (Tab. S3), following recommendations by Jochum et al. (2012) for low mass resolution. Dwell times were adjusted based on the natural abundance of the selected isotope within carbonates (Tab. S3). Most elements were analyzed in single quadrupole mode. In addition, to improve the limits of detection for Fe, the samples were analyzed by using triple-quadrupole technology with reaction cell. Considering the nature of the samples, the analysis of low Fe concentrations is compromised by the formation of <sup>40</sup>Ca<sup>16</sup>O polyatomic species which is significant during the analysis of Ca rich samples. To avoid such interferences, a reaction gas  $N_2O$  at a flow rate of approximately 0.4 - 0.5 mL/min mixed with He at a flow rate of 0.6 - 0.8 mL/min was introduced into the reaction cell between the two quadrupoles. The main isotope of Fe ( $^{56}$ Fe - 91.76%) is analyzed by mass shifting Fe to FeO at m/z = 72 (i.e. with the first quadrupole set to transmit m/z = 56 and the second quadrupole set to transmit m/z = 72). The glass reference material NIST612 was used as external standard, and GeoReM preferred values were used as the reference concentrations (Jochum et al., 2011). In addition, the nano powdered *Porites* sp. carbonate reference material (RM) JCp-1-NP was measured in each analysis block for quality control. <sup>43</sup>Ca was used as internal standard, and the Element/Ca (El/Ca) ratios accounted for varying laser ablation yields when measuring the carbonate (Rosenthal et al., 1999). Each analysis block consisted of approximately 30 analyses on various coral tips bracketed by two measurements of NIST612 and one measurement of JCp-1-NP. After analysing five points on a single tip, the tip was randomly switched to another to mitigate time dependent effects arising from potential fluctuations in plasma conditions. Outliers were determined based on the interquartile range (IQR) method: data points that fall below the lower quartile Q1 – 1.5\*IQR or above Q3 + 1.5\*IQR, with IQR calculated as the difference between Q3 and Q1, were removed from the dataset. Element specific limits of detection (LOD) were calculated according to Longerich et al. (1996) (6):

$$LOD = \frac{3\sigma_b}{S} \sqrt{\frac{1}{n_{bg}} + \frac{1}{n_a}},\tag{6}$$

with  $\sigma_b$  the standard deviation of the background counts, S the sensitivity, determined on the calibration material NIST SRM 612,  $n_{bg}$  the background counts, and  $n_a$  the analyte counts. If measured concentrations fell below the calculated LODs, those values were excluded from further data processing. Trace metal (V, Mn, Fe, Co, Ni, Cu, Cd, Pb) data are presented as El/Ca ratios [µmol/mol] or as mass fraction [mg kg<sup>-1</sup>], to allow the comparison with tissue and symbiont concentrations, as they cannot be effectively normalized to Ca. The transformation from [µmol/mol] to [mg kg<sup>-1</sup>] was based on 40.04 wt.% Ca and comes with an error due to the assumption that the measured skeleton consists solely of calcium carbonate, which in reality is a mix of Ca and various trace metals, which may affect the accuracy of the transformation. Additionally, alkali metals, alkaline earth metals (Li/Ca, B/Ca, Na/Ca and Mg/Ca), Sr/Ca, and U/Ca were measured in the coral skeleton as they are typically utilized for paleo reconstruction (e.g. Beck et al., 1992; Mitsuguchi et al., 1996; Wu et al., 2021).

Accuracy, reproducibility and internal consistency were assessed on the RM JCp-1-NP (myStandards GmbH). Quality control measures included comparisons of obtained results to published values by the manufacturer (myStandards GmbH) and to various LA-ICP-MS studies (Boer et al., 2022; Mccormick et al., 2023; Jochum et al., 2019) (Tab. S4). As highlighted in Jochum et al. (2019), the relative standard deviation (RSD) of elements within the natural reference material is strongly dependent on their abundance in the JCp-1-NP. Abundant elements such as Sr, Na and Mg exhibited a RSD of  $\leq$  4% and recovery rates (calculated as measured concentration over the reference concentration) approximating 108 % (Tab. S4). For metals present in low concentrations, RSD values generally remained below 13% (except for Cd) with most recoveries deviating  $\leq$  20% from the reference values (Tab. S4). Notably, Ni, Fe and Zn displayed lower average recoveries (57-78%), while V and Mn were consistently overestimated compared to published values (averaged recovery of 130% for V, and 155% for Mn) (Tab. S4). The observed variations in the recovery rates of trace metals within the JCp-1-NP can be partially attributed to the non-matrix matched calibration with the NIST 612 silicate matrix (Miliszkiewicz et al., 2015; Jochum et al., 2007), to the limited number of published trace metal data for JCp-1-NP, and the reported variability in trace metal content therein (Tab. S4). Even as accuracies occasionally deviated from the acceptable range (>  $\pm$ 20%), our methodology remained robust, anchored by the comparison of measured concentrations in the ash exposed skeleton to a control reference. This

approach provided a reliable basis for data interpretation, acknowledging the observed variability in the recovery of specific trace metals within the JCp-1-NP.

# 2.3.2 Apparent skeletal distribution coefficient $K_{El}$

Apparent distribution coefficients ( $K_{El}$ ) for control and ash exposed coral skeletons were calculated using El/Ca ratios of skeletons and the seawater, as outlined in Eq.(7) (Bruland, 1983):

$$K_{El} = \frac{El/_{Ca_{skeleton}}}{El/_{Ca_{seawater}}} \tag{7}$$

This calculation assumes that the calcifying fluid has the same composition as seawater.  $El/Ca_{skeleton}$  ratios were measured using LA-ICP-MS, while the El/Ca<sub>seawater</sub> ratios were calculated using seawater blank concentrations for the control samples. For ash exposed samples, the average of the modelled element concentrations in the culture tank was used (Fig. 3, Tab. S5). Seawater Ca concentrations were assumed to be unaffected by volcanic ash leaching, as leaching studies by Horwell et al. (2023) revealed a low Ca input from the La Soufrière ash compared to the high background Ca concentration in seawater. To determine  $El/Ca_{seawater}$ , we normalized the modelled trace metal concentrations to a constant seawater Ca concentration of 455 mg/L, which corresponds to measured Ca concentrations in Mediterranean Sea at  $\approx 38.5$  % (Fritzmann et al., 2007; Krumgalz and Holzer, 2003).  $K_{El}$  was determined for V, Mn, Fe, Co, Ni, Cu, Zn, Cd and Pb.

### 315 **2.4. Statistical Analysis**

310

320

325

Data were examined for normal distribution using the Shapiro-Wilk test and assessed for homoskedasticity via the Levene test. For normally distributed data, a single one-way Analysis of Variance (ANOVA) was conducted. In instances where the normality test was rejected, the nonparametric Wilcoxon-Mann-Whitney test (WMW) was performed. If Levene's test was rejected, but the data still exhibited normal distribution, the Welch t-test (Welch) was utilized. To account for potential tank effects on the chemical composition of coral skeletons, nested ANOVA and Tukey's post-hoc tests were performed. Statistically significant differences between groups were defined as p 

Fig. 3: Trace metal content [nmol L<sup>-1</sup>] in 0.22 µm filtered Mediterranean seawater before and after the addition of volcanic ash. The bar plot on a logarithmic scale shows results from the 24 h ash leaching at a 1 g: 100 mL ratio (0.3 g ash: 30 mL. seawater).

# The estimated trace metal concentrations in the coral culture tanks (orange) at a 1 g: 12 L ratio (2.5 g: 30 L) were modelled. Data for the ash leaching experiment is presented as mean concentration (n = 3) $\pm$ SD.

The metal concentrations, in [mg kg<sup>-1</sup> ash], obtained in this study are in agreement with results by Horwell et al. (2023), who measured similar leaching amounts of Fe, Mn and Ni from the La Soufrière (St. Vincent) ash after one hour at the same leachate ratio (Tab. 3). Although a different leaching solution (deionized water) was used, most metals were released at similar concentrations. However, their concentrations tended to be higher when deionized water is used (Jones and Gislason, 2008), likely due to pH differences in the leaching solutions (deionized water: ≈5.5, seawater at the CSM: ≈7.9-8.0). However, the metal input from La Soufrière ash is not representative of average ash leaching. Compared to the global median concentrations compiled by Ayris and Delmelle (2012), from a range of ash leachate analyses across different leachate ratios, the volcanic ash in our study consistently released lower levels of water extractable metals, except for Zn, which exceeded the global values by a factor of two (Tab. 2).

Table 2: Comparison of leached metal concentrations from the La Soufrière ash, collected on Barbados [mg kg<sup>-1</sup> ash]

| Element           | This study  | BB_01     | BB_02        | Global median          |  |
|-------------------|-------------|-----------|--------------|------------------------|--|
|                   | Tills study | DD_01     | DD_02        | concentrations         |  |
|                   | [mg/kg]     | [mg/kg]   | [mg/kg]      | [mg/kg]                |  |
| V                 | -0.010      | NA        | NA           | 0.089                  |  |
| Mn                | 4.808       | 8.1       | 1.8          | 20                     |  |
| Fe                | 0.125       | 

Fig.4: Trace metal concentrations in the three coral compartments (skeleton, tissue and symbionts) of *S. pistillata* in control (blue) and after 6 weeks of exposure to 250 mg L<sup>-1</sup> per week (orange). (a) Relationship between metal concentrations in the coral host tissue, the symbionts, and the coral skeleton. (b) Relationship between metal concentrations in the seawater (blanks and modelled values) and the coral skeleton. Elements are numbered as follows: 1–V, 2–Mn, 3–Fe, 4–Co, 5–Ni, 6–Cu, 7–Zn, 8–Cd, 9–Pb. (c) Partitioning of trace metals in tissues, symbionts and skeletons. All graphs are plotted on a logarithmic scale.

Metals more abundant in seawater, such as Zn (28 nmol  $L^{-1}$ ), were also found at higher concentrations within the coral (skeleton:  $\approx$ 0.3 mg kg<sup>-1</sup>; tissue:  $\approx$ 130 mg kg<sup>-1</sup>; symbionts:  $\approx$ 150 mg kg<sup>-1</sup>), indicating an exchange between the surrounding seawater and coral holobiont (Mitchelmore et al., 2007). The coral's extracellular calcifying fluid (ECF), from which the aragonite skeleton is precipitated, is directly connected to seawater through 20 nm wide paracellular pathways (Ganot et al., 2020). This connection allows the ECF to act as a semiclosed seawater reservoir (Ram & Erez, 2023), underscoring the strong influence of seawater chemistry on the trace metal distribution in coral aragonite skeletons. Supporting this, Schmidt et al. (2024) demonstrated through culturing experiments a positive correlation between skeletal trace metal concentrations (Mn, Ni, Zn, Cd and Zn) and their measured levels in the seawater. This dependency highlights the potential of coral skeletons as bioindicators of metal contamination (Ali et al., 2010; El-Sorogy et al., 2016; Nour and Nouh, 2020).

Although coral tissue has been proposed as good indicator of water quality (Esslemont et al., 2000; Esslemont et al., 2004; Runnalls and Coleman, 2003), with a rapid increase in metal concentration upon exposure (Metian et al., 2015), other invertebrates inhabiting the same contaminated environments tend to exhibit greater metal accumulation (Brown and Holley, 1982). Nevertheless, coral tissues and symbionts contained higher concentrations of the investigated metals than their skeleton (Fig. 5a) (Reichelt-Brushett and Mcorist, 2003), with especially Fe and Zn exhibiting a strong enrichment in the soft tissues (in both coral host and symbionts) compared to the skeleton (Anu et al., 2007a). While Co tended to be preferentially accumulated in the coral tissue (Fig. 4c), the majority of analyzed metals (V, Fe, Cu and Zn) appeared to be equally concentrated in tissue and symbionts. Although trace metal partitioning among coral compartments is metal specific (Esslemont et al., 2000), the tissue/skeleton and symbionts/skeleton trace metal ratios remained stable independent of experimental condition (Fig. 4c). Therefore, constant metal quotas in coral host tissue and symbionts appear to be internally controlled, as the metal input from volcanic ash led to nonuniform changes in their tissue/skeleton and symbionts/skeleton ratios, which largely remained within the margin of error of the control condition (Fig. 4c). To sustain productivity over a limited period, corals actively farm and consume their photosynthetic symbionts (Wiedenmann et al., 2023; Titlyanov et al., 1996), with 1 to 6 % of symbionts degrading daily in S. pistillata (Titlyanov et al., 1996). The digestion of excess symbiont cells has been shown to alter macronutrient content in the coral host, suggesting that the same alteration can be expected as response to trace metal enrichment. This would lead to changes in the host metallome to maintain metal specific needs.

# 3.2.2 Geochemical signals of volcanic ash leaching in coral compartments

- To assess potential metal supply following ash addition and determine which coral compartment's metallome is most impacted, we used the branching coral *S. pistillata*. This species has been proposed as a biomonitoring tool (Nour and Nouh, 2020), due to its large reactive skeletal surface area and fast growth rate (Kotb, 2001; Liberman et al., 1995; Shaish et al., 2006). Control and ash exposed corals were reared in similar culture conditions, with previous findings confirming that the physical effect of ash addition (i.e. a decrease of 16 ± 9 % in photosynthetically active radiation, which returned to initial culture conditions after 2-4 hours, and no detectable change in monitored pH) was negligible (Förster et al., 2024). The primary difference between the experimental conditions was in the chemical composition of the seawater, which was slightly altered by volcanic ash addition (Fig. 3). Our approximated trace metal concentrations in the seawater (Tab. S5), derived from the compartment model, indicate a negligible effect of ash exposure compared to seawater blanks, except on Mn and Co concentrations.
- No significant differences were observed in tissue and symbiont metal concentrations between tanks maintained under the same condition (p > 0.05 in all cases, except Ni in the control tissues; Tab. S7a&b). The addition of ash generally did not result in a significant increase in the mean metal concentrations in coral tissue or symbionts (Fig. 5a, Tab. S8). While coral tissues and symbionts exhibit a dynamic range and higher metal concentrations than the skeleton (Fig. 4), no significant enrichment compared to the control condition was observed, except for higher Mn (Welch, p < 0.006) and Fe (ANOVA, p < 0.03) concentrations in the ash exposed coral tissue. In contrast, the metal concentrations in coral symbionts remained

unaffected by volcanic ash (p > 0.5 for all metals; Fig. 5a, Tab. S8).

The presence of volcanic ash changed the composition of the coral skeleton significantly as the majority of analyzed metals (V, Mn, Fe, Ni, Zn), except Co and Cd, showed a significant difference between the two experimental conditions (Fig. 5b). Trace metal concentrations were generally higher in ash exposed coral skeletons compared to the control, marked by a 2.4-, 2.1-, 1.26-fold increase in Fe, Mn and Ni concentrations, respectively (all: WMW, p  $\ll$  0.001). Zn and V concentrations in the ash exposed skeletons were 1.15- and 1.12-times higher than in the controls (both: WMW, p  $\ll$  0.002). In contrast, Pb and Cu concentrations were both (0.65- and 0.9-times) lower in ash exposed skeletons and significantly more concentrated in control samples (WMW, p  $\ll$  0.001 for Pb; p 

Fig. 5: Trace metal content of *S. pistillata* in control (blue) and after 6 weeks of exposure to 250 mg L<sup>-1</sup> per week (orange). (a) Trace metal concentrations [mg kg<sup>-1</sup>] in the three coral compartments (skeleton, tissue and symbionts). Skeleton data comprises 360 measurements on 12 coral apices, and 390 measurements on 12 ash exposed coral apices. Tissue and symbionts from 12 coral nubbins (n = 6 per condition) were analysed. (b) Trace metal concentrations in control and ash exposed skeletons. Data are presented as El/Ca [μmol/mol] to account for ablated carbonate mass. The white point represents the mean value, the black line the median value, the black box delimits the 1st and 3rd quartile, and the whiskers give the range within 1.5-times the Inter-Quartile Range (IQR, defined as Q3-Q1) from the quartiles. The width of the violin plots indicates data density. Statistically significant differences between treatments are represented by asterisks (\*), with an indication of the level of significance in the number of asterisks.

Ash addition led to no measurable changes in the trace metal composition of the coral tissue (except for Mn and Fe) and symbionts, but showed significant increases in the coral skeleton. To quantify the ability of the coral to monitor chemical changes, we used a relative enrichment factor (rEF), which compared the mean metal concentration in ash exposed compartments to that of the controls (rEF<sub>coral compartment</sub>) (Fig. 6). A rEF > 1 indicates a metal enrichment in the ash exposed condition, while rEF 

Fig. 6: Relative enrichment of trace metals in individual coral compartments (skeleton, tissue and symbionts) of the coral S. pistillata after volcanic ash exposure. (a) Relationship of rEF<sub>coral compartments</sub> and trace metal content in seawater. Data are presented as relative enrichment factor (rEF), calculated as the mean concentration of the El [mg kg<sup>-1</sup>] in the ash exposed condition/ mean concentration El [mg kg<sup>-1</sup>] in the control condition. A rEF > 1 indicates an enrichment in the ash exposed compartment, while a rEF < 1 showcases a depletion. (b) Relationship of rEF<sub>skeleton</sub> and rEF<sub>seawater</sub>. The rEF<sub>seawater</sub> is calculated using the modeled trace metal concentrations in the culture tanks and comparing it to the blank seawater values. Color corresponds to the trace metal concentration in the skeleton. All graphs are plotted on a logarithmic scale.

The  $rEF_{skeleton}$  did not correlate with the enrichment observed in the ash exposed seawater ( $rEF_{seawater}$ ) (Fig. 6b). For example, the 5-fold increase in Mn and 2-fold increase in Co concentrations in the ash exposed seawater did not match the concentrations in the skeleton (i.e., out of the 1:1 line; Fig. 6b), suggesting that leaching rates exceeded the metal uptake rate of the coral. Thus, trace metal accumulation in the skeleton did not seem to be primarily driven by the metal availability in the seawater, but is more likely an expression of the organisms internally regulating metal quotas, except for when

concentrations vary across several orders of magnitude (Fig. 4a). Several pathways have been proposed how element incorporation into the skeleton can occur: (i) the replacement of Ca<sup>2+</sup> with similarly sized and charged cations in the 490 aragonite crystal lattice (Livingston and Thompson, 1971; Sholkovitz and Shen, 1995), (ii) the binding with the intraskeletal organic matrix or other endolithic microorganisms, (iii) attachment on crystal distortions and structure defects (Mitsuguchi and Kawakami, 2012; Montagna et al., 2014) and (iv) adsorption of metals on tissue-exposed skeleton (Brown et al., 1991) Many essential elements, such as Fe, are important cofactors in various metabolic and enzymatic processes (Reich et al., 495 2020: Reich et al., 2023), undergoing a dynamic uptake to fulfill the fluctuating demands for essential metals. Synergistic and antagonistic interactions between multiple trace metals, as demonstrated for certain cultures of Symbiodiniceae (Reich et al., 2020), can overcome the limitation of another metal. Hence, metal uptake in coral tissue and symbionts is largely affected by these varying metal requirements. The Fe and Mn enrichment observed in both tissue and skeleton (Fig. 5a), might indicate an increased metabolic demand. Another potential explanation for the deviation from the 1:1-line in rEF<sub>skeleton</sub> 500 and rEF<sub>seawater</sub> is the method used to calculate rEF<sub>seawater</sub>. While geochemical signals measured in specific coral compartments are reliable, trace metal concentrations in coral culture tanks were estimated based on the quantitative addition of leached metals from the leaching study (section 2.2.1). However, assuming mean concentrations over the six-week experimental period (Tab. S5), may not accurately reflect the actual tank concentrations, given the complex metal specific leaching profiles (Fig. 1b) resulting from the thrice-weekly addition of 2.5 g volcanic ash.

It has to be tested whether the arithmetic mean is a reasonable approximation for the pulsed, high intensity metal input that affects the tank concentration due to ash addition. Furthermore, metabolic changes may influence metal uptake and organismal trace metal partitioning. The transfer and sequestration of metals into the skeleton occurs during calcification (Metian et al., 2015), a process that is enhanced by light (Gattuso et al., 1999). During the day, ash derived metal concentrations peaked, followed by a rapid decline once the seawater inflow was restored. Radiotracer studies by Metian et al. (2015) have shown that *S. pistillata* can retain metals in its tissue for over 40 d, despite continuous translocation into the skeleton. Further research is needed to understand how pulsed supplies of essential metals affect uptake and translocation kinetics during the day and over time.

### 3.2.3. Trace metal distribution within the coral skeleton

Analysis on 360 LA-ICP-MS point measurements on 12 control coral apices and on 390-point measurements on 12 ash exposed coral apices revealed differences in trace metal correlations between the two experimental conditions (Fig. 7).

Fig. 7: Trace metal distribution in the coral skeletons of *S. pistillata* in control and after 6 weeks of exposure to volcanic ash. (a) Correlation matrix of trace metal concentrations within the control (upper part) and ash exposed (lower part) skeletons. Colour and size of squares correspond to the calculated Pearson correlation coefficient and r values range from -1 (strong negative correlation) to 1 (strong positive correlation). (b) Relationship between skeletal Ni/Ca, Mn/Ca, Pb/Ca, Cu/Ca, Mg/Ca and U/Ca ratios with 95% confidence ellipsoids for each condition (control-blue, ash exposed-red). Squares resemble individual measurements on control apices, and triangles correspond to measurements on the ash exposed coral tips. A third El/Ca ratio is utilized as colour code from low concentration (red fill) to high concentration (blue fill) of the respective ratio. If the symbol does not contain a coloured fill, this indicates that the measured concentration is below the limit of detection.

Corals exposed to volcanic ash showed a tendency towards stronger positive correlations within the transition metal content (Fig. 7a). For example, Mn/Ca and Ni/Ca, exhibited a significant positive correlation in ash exposed skeletons (r = 0.58, p < 0.001) in contrast to the controls (r = 0.35, p = 0.33). Another key difference between ash exposed and control skeletons was the transition metal behavior in relation to alkali and alkaline earth metal concentrations, such as Li/Ca, Na/Ca and Mg/Ca (Tab. S10). In both experimental conditions, we observed similar strong positive correlations between Na/Ca, Li/Ca and Mg/Ca ( $r = \approx 0.7$ , p < 0.0001 between each; Fig. 7a & Tab. S10) (also observed in Rollion-Bard and Blamart (2015)), and all three El/Ca ratios (Li/Ca, Na/Ca, and Mg/Ca) displayed weak positive correlations to transition metals (Mn, Ni, Cu and Zn) in the controls (Tab. 3 & S12), except for Ni/Ca which correlates significantly with Li/Ca (r = 0.48, p < 0.009), Na/Ca (r = 0.38, p < 0.02) and Mg/Ca (r = 0.33, p < 0.03). In contrast, ash exposure reversed this by producing weak negative correlations (Tab. 3 & S12).

Table 3: Correlation analysis of several El/Ca ratios with Li/Ca, Na/Ca and Mg/Ca in the skeletons of *S. pistillata*. Correlation strength is presented as Pearson correlation coefficient (r) and statistically significant correlations exists when p < 0.05.

| El/Ca ratio           |       | Positive correlation |            | Negative correlation |             |             |             |
|-----------------------|-------|----------------------|------------|----------------------|-------------|-------------|-------------|
|                       |       | Li                   | Na         | Mg                   | Li          | Na          | Mg          |
| Control skeletons  O  | V     | (r = 0.03,           | (r = 0.03, |                      |             |             | (r = -0.11, |
|                       | V     | p = 0.21)            | p=0.20)    |                      |             |             | p = 0.13)   |
|                       | Mn    | (r = 0.10,           | (r = 0.13, | (r = 0.04,           |             |             |             |
|                       | IVIII | p = 0.82)            | p = 0.82)  | p = 0.72)            |             |             |             |
|                       | Co    |                      |            |                      | (r = -0.23, | (r = -0.18, | (r = -0.36, |
|                       | Co    |                      |            |                      | p < 0.02)   | p < 0.02)   | p < 0.008)  |
|                       | Ni    | (r = 0.48,           | (r = 0.38, | (r = 0.33,           |             |             |             |
|                       | 141   | p < 0.009)           | p<0.02)    | p < 0.02)            |             |             |             |
|                       | Cu    | (r = 0.26,           | (r = 0.34, | (r = 0.25,           |             |             |             |
|                       | Cu    | p = 0.13)            | p=0.10)    | p = 0.13)            |             |             |             |
|                       | Zn    | (r = 0.23,           | (r = 0.26, | (r = 0.27,           |             |             |             |
|                       | p =   | p = 0.11             | p=0.10)    | p = 0.09)            |             |             |             |
|                       | Pb    |                      |            |                      | (r = -0.12, | (r = -0.12, | (r = -0.25, |
|                       | 10    |                      |            |                      | p < 0.005)  | p < 0.005)  | p < 0.002)  |
| Ash exposed skeletons | V     | (r = 0.03,           | (r = 0.08, |                      |             |             | (r = -0.07, |
|                       |       | p = 0.32)            | p = 0.35)  |                      |             |             | p = 0.26)   |
|                       | Mn    | Mn                   |            | (r = 0.10,           | (r = -0.03, | (r = -0.01, |             |
|                       | 14111 |                      |            | p = 0.41)            | p = 0.30)   | p = 0.80)   |             |
| exp                   | Co    |                      |            |                      | (r = -0.16, | (r = -0.04, | (r = -0.09, |
| Ash                   | Co    |                      |            |                      | p = 0.26)   | p = 0.29)   | p = 0.37)   |
|                       |       |                      |            |                      |             |             |             |

| Ni  | (r = 0.02, | (r = 0.18, |             | (r = -0.01, |              |
|-----|------------|------------|-------------|-------------|--------------|
| INI | p = 0.82)  | p = 0.62)  |             | p=0.74)     |              |
| Cu  |            |            | (r = -0.13, | (r = -0.1,  | (r = -0.24,  |
|     |            |            | p < 0.03)   | p < 0.02)   | p < 0.02)    |
| Zn  |            |            | (r = -0.12, | (r = -0.07, | (r = -0.08,  |
|     |            |            | p = 0.17)   | p = 0.12)   | p = 0.28)    |
| Pb  |            | ·          | (r = -0.32, | (r = -0.23, | (r = -0.49,  |
|     |            |            | p < 0.0007) | p < 0.003)  | p < 0.00005) |

Li/Ca, Na/Ca, and Mg/Ca are often utilized as proxies for paleoclimate reconstruction (e.g. Beck et al., 1992; Mitsuguchi et al., 1996; Wu et al., 2021), and higher Mg/Ca ratios were linked to higher ratios of other metals in biogenic aragonite (Ulrich et al., 2021). In abiogenic aragonite precipitation experiments, increasing element concentrations such as Ni and Co were linearly correlated with mineral growth rates (Brazier and Mavromatis, 2022). In biogenic aragonite this relationship is not well established, and no proxy exists for coral growth rate. Nevertheless, certain El/Ca ratios appeared to be influenced by 545 skeletal growth rates, such as Mg/Ca, which was shown to correlate with calcification rates of the branching coral Acropora sp. (Bell et al., 2017; Reynaud et al., 2007). Since Mg<sup>2+</sup> is too small to stably occupy the Ca<sup>2+</sup> vacancy in an aragonite lattice (Goldschmidt, 1954), it is adsorbed onto crystal discontinuities or incorporated into lattice defects, as proposed for K<sup>+</sup> and Na<sup>+</sup> (Mitsuguchi et al., 2010). Likewise, Li<sup>+</sup> is unlikely to replace Ca<sup>2+</sup> in the aragonite due to its ionic size and monovalent charge, requiring either internally regulated charge balance via the coupled substitution with a second trace metal or the 550 creation of lattice vacancies. The strong positive correlation between Li/Ca and Mg/Ca in the coral skeleton (Fig. 7, with r = 0.83, p < 0.0001 in control; r = 0.77, p < 0.0001 in ash exposed skeletons), suggests a similar uptake mechanism. A direct paracellular seawater transport to the ECF for both elements is proposed (Montagna et al., 2014). Rollion-Bard and Blamart (2015) show strong correlations between Na/Ca, Li/Ca and Mg/Ca ratios in three scleractinian corals, supporting a unified incorporation mechanism into the aragonite skeleton. They propose that these elevated ratios may indicate an enhanced coral 555 growth rate. Given the limited impact of volcanic ash leaching on alkali and alkaline earth element concentrations in seawater, owing to their already high initial levels (e.g. Millero et al., 2008), it is reasonable to associate increased Li/Ca, Na/Ca and Mg/Ca ratios with enhanced growth (Rollion-Bard and Blamart, 2015). From an abiotic perspective, faster growth rates result from a higher argonite saturation following the rate law of calcium carbonate formation  $R = k^*(\Omega - 1)^n$ with R being the precipitation rate,  $\Omega$  the aragonite saturation, k the rate constant and n the order of reaction. It can be seen 560 that R is directly proportional to  $\Omega$  (e.g. Burton and Walter, 1987). In corals an increase in saturation state can be achieved by stronger proton-Ca exchange and active bicarbonate transport into the coral CF (Mcculloch et al., 2017; Sevilgen et al., 2019). If Ca enrichment in the CF would be the only cause, it would lead to a decrease rather than increase in these ratios during phases of high growth rate, due to dilution. As this is not observed, either defect site incorporation is overwhelming the Ca enrichment effect or Ca gets depleted during phases of high precipitation, while the bicarbonate transport maintains a

high  $\Omega$  until the CF is again refreshed with seawater.

Volcanic ash exposure induced significant physiological changes in S. pistillata, leading to elevated skeletal growth rates over six weeks (Förster et al., 2024). However, we did not observe positive correlations between Li/Ca, Na/Ca and Mg/Ca and the transition metal content in ash exposed coral skeletons (Tab. 3). This can be partially attributed to the observed large variability in skeleton metal composition within one experimental condition (Fig. 7b & S7b), which is evident in the generally low to medium Pearson coefficients (Tab. S10). Growth rate and other physiologically processes are believed to be the primary driver of geochemical heterogeneities within branching coral skeletons, while environmental changes play a secondary role (Shirai et al., 2008). Physiological responses are likely affecting the geochemical signatures during the formation of the coral skeleton ("vital effects", Urey et al., 1951), but remain poorly constrained. Coral skeletogenesis is biologically regulated (Von Euw et al., 2017; Tambutté et al., 2011), resulting in a composition biologically controlled (Meibom et al., 2004; Stolarski et al., 2021) and a structure influenced by cellular processes (Coronado et al., 2019). Numerous studies have documented the compositional heterogeneity within coral skeletons (i.e. Esslemont et al., 2000: Meibom et al., 2008; Schmidt et al., 2024; Rollion-Bard and Blamart, 2015). For example, studies on Siderastrea siderea have shown that while coral geochemistry varies between different skeletal elements, it remains relatively consistent within the same type of skeletal structure (Galochkina et al., 2023; Chalk et al., 2021). To minimize chemical variability in our samples we focused on one particular anatomical feature, the theca. Each skeletal feature is composed of two distinct structural components: the organic-rich centers of calcification (COCs), which initiate biomineralization, subsequently overgrown by aragonite fibers (Nothdurft and Webb, 2006; Cuif and Dauphin, 2005; Stolarski, 2003). COCs are finely distributed within the theca of S. pistillata with sizes ranging from approximately 5 to 10 µm (Von Euw et al., 2017). COCs are chemically distinctively different from the surrounding aragonite fibers, as they have been shown to be anomalously enriched or depleted in certain trace metals (Meibom et al., 2006; Standish et al., 2024; Gagnon et al., 2007; Robinson et al., 2014). Another potential source for trace metal heterogeneities on a microscale can be attributed to the skeletal thickening process, which increases skeletal density with distance to the coral tip in branching corals (Shirai et al., 2008). Secondary precipitated aragonite has been found to exhibit a different trace metal signature. Each LA-ICP-MS point measurement provided an averaged concentration that reflected a mixture of sampled COCs, fibrous aragonite, primary formed and infilling aragonite within the 60 µm radial spot size. The relative proportions of these phases within each spot are unknown. To minimize sampling bias when comparing individual coral tips with each other, we analyzed and compared the overall geochemical pattern across multiple samples within each experimental condition. Changes in the skeletal trace metal distribution between experimental conditions might therefore result from cumulative effects of higher seawater metal availability through ash leaching, enhanced coral physiological processes due to the alleviation of micronutrient limitations, and a consequence of the laser ablation spot size during the ICP-MS measurement.

# 3.2.4. Apparent skeletal distribution coefficients of cultured S. pistillata

The patterns of the apparent distribution coefficients  $K_{El}$  showed differences between the cultured coral samples reared in control and ash exposed conditions (Fig. 8). In both conditions, Pb/Ca and Cd/Ca exhibited a higher incorporation in the skeleton compared to the seawater ( $K_{Pb} > 3.6$ ,  $K_{Cd} > 1.2$ ), while only control Co/Ca was higher in skeletons than in the seawater ( $K_{Co} > 1.2$ ). In contrast, V/Ca, Fe/Ca, Ni/Ca, Zn/Ca, Cu/Ca, and Mn/Ca showed higher availability in seawater, than in the coral skeleton ( $K_{El} 

Fig. 8: Partitioning of trace metals between seawater and coral skeleton of *S. pistillata* in the control condition (blue) and after 6 weeks of exposure to 250 mg L<sup>-1</sup> per week (orange). (a) Relationship of El/Ca<sub>skeleton</sub> and El/Ca<sub>seawater</sub>. Distribution coefficients *K<sub>El</sub>* are plotted as diagonal dotted lines (grey). Ca concentrations in seawater are assumed to be constant in both experimental conditions and were set as 455 mg/L (Krumgalz and Holzer, 2003). Data are presented as mean concentration and plotted on a logarithmic scale. Error bars for El/Ca<sub>seawater</sub> are calculated using the RSD from triplicate seawater blank measurements, while error bars for El/Ca<sub>skeleton</sub> are based on the RSD from JCp-1-NP. (b) Relationship of *K<sub>El</sub>* and the seawater metal concentration [nmol L<sup>-1</sup>].

615 In this study, we present -to our knowledge for the first time- skeletal distribution coefficients for cultured S. pistillata. Our findings reveal, a general decrease of  $K_{El}$  with increasing metal concentration in the seawater (Fig. 8b). A similar response was observed in skeletons from cultured *Porites* sp. (Schmidt et al., 2024), which exhibited decreasing K<sub>El</sub> of trace metals (Cr, Mn, Ni, Zn, Ag, Cd and Pb) with increasing seawater El/Ca ratios. Most El/Ca in their skeleton showed a nonlinear response to seawater El/Ca enrichment, except for Pb/Ca, which showed a robust linear response to changes in seawater 620 Pb/Ca. Ash exposure slightly increased the metal availability in the seawater and affected specific  $K_{El}$ .  $K_{El}$  are not constant and were affected by an increase in the trace metal availability (Fig. 8), and presumably by the related physiological effects (Förster et al., 2024). Ulrich et al. (2021) reported that  $K_{El}$  of the biogenic aragonite in corals is generally lower than the  $K_{El}$ of abiogenic precipitated aragonite, highlighting a lower trace metal incorporation into the coral skeleton compared to inorganically grown crystals. This difference in  $K_{El}$  suggests that corals (and their dinoflagellates) actively regulate the 625 uptake of specific trace metals to maintain basic metabolic functions (Rodriguez et al., 2016). However, beyond a certain metal threshold, toxic effects might occur, triggering biochemical mechanisms that can expel excess metals (e.g. through the expulsion of symbionts (Meehan and Ostrander, 1997; Peters et al., 1997). For most measured trace metals, we observed significant increases in their skeleton El/Ca ratios (Fig. 5b and Tab. S8), while the seawater El/Ca remained largely unaffected (Fig. 3). This could indicate that corals might preferentially incorporate toxic metals into their growing skeleton 630 as part of a proposed metal detoxification mechanism (Metian et al., 2015; Anu et al., 2007b). A contributing factor to the measured differences in calculated  $K_{El}$  between ash exposed and control conditions might be the intra- and inter-colony geochemical heterogeneities in the coral skeletons (i.e. Schmidt et al., 2024; Meibom et al., 2008; Shirai et al., 2008), which we tried to minimize (as explained above).

# 5. Conclusions

We investigated the geochemical impact of volcanic ash leaching on the metallome of the hermatypic coral *S. pistillata*. Ash leaching experiments revealed, that volcanic ash from La Soufrière rapidly released trace metals (Mn, Zn, Co, Cu, Cd, Fe and Ni) into seawater, while Pb was removed from the solution. However, the overall leaching potential of the ash was below the global average, except for Zn. Based on these results, we developed a compartment model to estimate the real-time metal concentrations in coral culture tanks, in which corals were reared under control and ash exposure for six weeks.

Except for Mn concentrations being, on average, five times higher and the Co concentration twice as high in the ash exposed tanks, no metal enrichment was calculated. However, trace metal concentrations generally varied significantly among the three distinct coral compartments, with higher metal concentrations in tissue and symbionts than in the skeleton. Despite this variability, trace metal concentrations in all coral compartments correlated with seawater metal concentrations. Additionally, skeleton trace metal concentrations were linked to metal levels in both tissue and symbionts, regardless of exposure condition. Volcanic ash leaching did not affect trace metal concentrations in coral tissues (except for Mn and Fe) and symbionts, but led to notable changes in coral skeletons. Specifically, skeletons from ash exposed corals showed metal

enrichments for V, Mn, Fe, Ni, and Zn, along with a depletion of Cu and Pb compared to control corals. The internal trace metal distribution within the coral skeleton shifted in response to ash exposure, altering correlations between transition metals and alkaline earth/alkali elements, likely due to modified physiological processes. For the first time, skeletal distribution coefficients ( $K_{El}$ ) were calculated for lab grown S. *pistillata*, revealing a dependency on seawater metal concentration. Most trace metals were more abundant in seawater than in the coral skeleton ( $K_{El}$ <1), but the opposite was observed for Pb, Cd, and Co ( $K_{El}$ >1). This multielement coral culture study, highlights how volcanic ash influences trace metal dynamics by both supplying and scavenging essential and nonessential metals, altering seawater concentrations and reshaping trace metal cycling within the coral holobiont. Although no definitive skeletal proxy for volcanic eruptions was identified, the metallomic shifts observed in coral skeletons in response to ash-derived metal inputs, and calculated metal distribution coefficients, may help interpret geochemical signals in corals near volcanic islands. Further work is still needed to understand whether the metal content of coral skeletons scales with ash-loading in coastal environments and whether factors such as biological stress and oceanographic dynamics influence metal dissolution and uptake.

### **Author contribution**

650

655

670

FF is the primary author and contributed to all aspects of the manuscript. FF prepared the manuscript with contributions from all co-authors. TS and SF developed the model code, while FF performed the simulations. SF and AT contributed to method validation, review and editing. LS and EA conducted analyses and helped in the review and editing process. SR and CFP contributed to supervision, idea conceptualization, experimental design and editing. TS helped with supervision, idea conceptualization, results analysis, data processing, review and editing.

### 665 Acknowledgements

The authors express their gratitude to Cécile Rottier and Maria-Isabelle Marcus for their laboratory assistance during FFs time at the Centre Scientifique de Monaco (Monaco). We also appreciate John B. Mwansa (Barbados WaterAuthority) for collecting and supplying the volcanic ash samples used in the coral culture experiments. FF, SF and TS were funded by the Swiss National Science Foundation for the GEOVOLCO project (SNSF; Project no. PCEFP2\_194204). Additionally, SF was by the Swiss National Science Foundation (project 200020\_201106). This research benefits of the ClerVolc Program of Excellence of the International Research Centre of Disaster Science and Sustainable Development. » This study has been cofunded by CORDAP (Coral Research and Development Accelerate Platform) on the project: "Super supplement: boosting coral resilience through nutritional supplements.

# **Code Availability Statement**

The code for compartment model described in Sect. 2.2.1 of the current study is available from the corresponding author upon request.

### **Data Availability Statement**

The original data that support the findings of this study are available in Mendeley Data with the identifier doi:10.17632/c6dfb3gwv8.1.

### 680

710

### **Competing interests**

The authors declare that they have no known competing financial interests or personal relationships that could have appeared to influence the work reported in this paper.

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
