# Peer review of "Volcanic ash leaching alters the trace metal distribution within the coral holobiont of *Stylophora pistillata"

_EGUsphere, 2025_

## Author Comment (AC2)

**Response to the Reviewers**

**Anonymous Referee #1:**

The work by Förster et al. investigates the effects of volcanic ash exposure on corals across three biological compartments—tissue, skeleton, and symbiont. Studies that address all three compartments simultaneously are rare, and this manuscript sets a valuable precedent for improving our understanding of how changes in elemental composition affect coral physiology at different levels. The topic is novel and important: volcanic ash effects on corals remain understudied, and this work may also have broader implications for understanding how other disturbances introducing metals into marine environments impact coral health. The manuscript is generally well written, and the main ideas are easy to follow. However, I offer the following comments and suggestions to help improve the clarity and impact of the study.

The authors thank the referee for their feedback! The authors responded point-by-point in **Part I.** to the General Comments, and in **Part II.** to the Specific comments.

*Part I. Response to the General Comments*

Given the size and mixing limitations of the experimental tanks, how relevant are the ash concentrations and dissolution rates studied here to natural reef environments? Most reefs are found at depths starting around 1.5 m and extending much deeper, which differs considerably from the experimental setup.

There is not a clear biogeochemical justification for the specific amount of ash chosen, as the quantity and frequency of ash deposition can vary greatly, depending on eruption type, duration, and distance from a volcano. For example, the explosive phase from La Soufrière volcano lasted 13 days in 2021 (Cole et al., 2023), while in 1997 two explosive phases at Soufrière Hills Volcano (Montserrat) lasted 8 and 30 days. Both eruptive phases are shorter than the median eruption duration of 7 weeks (Siebert et al., 2015). During the 1997 eruptive periods of Soufrière Hills Volcano the interval between explosions varied between 2.5 and 63 h (Druitt et al., 2002), our experimental ash addition, three times a week, fall well within this range.

The dosage (250 mg L$^{-1}$ per week) was based upon typical sedimentation rates for reefs recorded in Barbados (a neighbouring island), but the goal of our study was not to replicate a single specific eruptive scenario. We tried to achieve a mechanistic understanding of the effects of dilute ash concentrations (limited effect on turbidity and no polyp smothering) on the trace metal distribution within corals. This is explained in lines 157-159 in the revised manuscript.

The discussion currently lacks a section addressing the possible mechanisms for trace element incorporation (e.g., V, Mn, Fe, Co, Ni, Cu, Zn, Cd, and Pb) into the coral skeleton. A consideration of these mechanisms would help interpret the measured concentrations and their biological significance.

This study primarily focuses on the metal uptake by metabolic demand, as well as metal accumulation and storage in the coral tissue and symbionts. While translocation of these metals into the skeleton is considered, the authors agree that multiple other pathways may also contribute to skeletal elemental incorporation. For this reason, a small paragraph summarising several proposed mechanisms for elemental incorporation into coral skeletons has been added (lines 480-484). The authors kept the

discussion rather broad, and did not differentiate individual metal pathways, as it is likely that a combination of these mechanisms act at the same time.

**Part II.** *Response to the Specific Comments*

Line 59 – Could you clarify why the lack of correlation is interpreted as evidence that metal release occurs via dissolution of metal salts? Could alternative processes, such as scavenging or desorption, also be playing a role?

The authors agree that the subject referenced in line 59 was not entirely clear. The authors argue, that due to the lack of correlation between the bulk and released Fe content, where low Fe ashes typically leach more Fe than high Fe ashes (Olgun et al., 2011), the leaching potential is likely dominated by rapid surface reactions, instead of a slow release of structural Fe. The key mechanism for volcanic ash Fe input in seawater involves metal salt dissolution, but ion adsorption and scavenging may also play a role. We revised the sentence for clarity (lines 62-66).

Line 70 – Please note that the cited source is not a peer-reviewed publication.

The Master's thesis by Jan-Christopher Fischer (2023) can be referred to as "grey" literature as there is no alternative from the formal literature. His work is online available and freely accessible on the University of Bristol website . The ash fall projections in his work are crucial for answering questions on how much and in which frequency corals in the coral triangle might experience varying layer thicknesses of deposited volcanic ash. Because this study is one of its kind, the authors believe it is justified to cite the work despite it not being a peer-reviewed publication.

Line 73 – The sentence is difficult to follow; adding commas may help clarify the meaning.

Lines 77-80: The sentence has been rewritten for clarity.

Line 77 – The phrasing here could be misleading, potentially implying that corals photosynthesize. Please reword for clarity.

Lines 82-84: "photosynthesis" has been removed, and "symbiont photosynthesis" has been added in the following line for clarity.

Line 97 – The term "trace metals" can be ambiguous in coral research, as trace metal concentrations in seawater and coral skeletons are not equivalent. It may be helpful to clarify this to avoid confusion.

Line 104: The sentence refers to the intraorganismal trace metal distribution of *Stylophora pistillata,* meaning we are referring to trace metals within the coral. For clarification we named the measured elements. Based on this comment we changed the following sentence (lines 104-106).

Line 118 – What is the basis for the target concentrations selected for the experiment?

A commonly used ash-to-leachate ratio in ash-leaching experiments is 1 g to 100 mL, which allows the assessment of water-extractable ions released from ash into the leachate (summarised in Stewart et

al., 2020). In our experiments, we obtained this ratio by mixing 30 mL of freshly collected 0.2 µm filtered seawater with 0.3 g of pristine volcanic ash in a 50 mL centrifuge tube. This reduced ash and seawater quantity was selected to minimize the material use while ensuring adequate mixing within the centrifuge tube on a rocker.

Line 150 – Same as above: what criteria were used to determine this specific concentration?

Please refer to the answer in **Part I.**

Line 152 – How was this statement assessed or quantified?

These lines refer to the effect of ash enrichment on shading and if it causes polyp smothering and how it was quantified. This study is an expansion of our previously published work (Förster et al. (2024), *STOTEN*), in which we documented the physiological coral response to ash exposure. Both studies originate from the same experiment, but while the present work focuses on geochemical cycling, the physiological aspects have been published previously and provide a valuable addition to the overall understanding. In the previous publication we also measured the photosynthetic active radiation (light intensity) within the tanks (which can be found as Figure A2 in the appendix of Förster et al., 2024). To quote section **3.1. Abiotic environmental parameters**: "[…] turbidity decreased PAR by 16 ± 9 %, and also minorly affected the irradiance on the neighboring control tanks. Suspended ash particles subsequently sedimented, so that the initial light intensity of 200 ± 10 µmol photons $m^{-2}$ $s^{-1}$ was restored 2-4 hours after the ash addition".
Polyp smothering was not observed macroscopically and biologically, as certain coral stress responses (coral bleaching, Lipid peroxidation or photosynthetic yields) were not observed.
For clarity we cited our previously published work (line 158).

Line 193 – Was the observation based on visual inspection or another method?

The weekly tank cleaning procedure involves the transfer of the coral nubbins to a different tank of the same condition, the complete water drainage of the tank to be cleaned, and the manual removal of all residual ash. The corresponding author, who performed the cleaning, ensures that all visible ash deposits were removed from the tanks.

Line 219 – There appears to be some redundancy between lines 219 and 220 (e.g., the lab is introduced twice).

Changed.

Line 226 – How confident are you that all organic material was removed? In my experience, tissue remnants can be difficult to eliminate entirely, even after thorough cleaning. Also, were the same tips used for airbrushing and LA-ICP-MS analysis?

We are confident that the superficial organic material was sufficiently removed. As all coral nubbins were airbrushed at the end of the experiments. Prior to embedding the ground coral apices in epoxy, we followed a bleaching protocol outlined in Tanzil et al. (2019), as detailed in lines 235-237. The bleaching, however, did not remove the intercrystalline organics.

Regarding the second points, there is a large overlap between the analysed apices and the corresponding tissue and symbiont data, but we have processed much more coral skeleton apices than tissue samples. This is because some of the airbrushed tissues were allocated for physiological and oxidative stress analyses, as presented in Förster et al. (2024), *STOTEN*.

Line 271 – The choice of references is somewhat odd. Galochkina focuses on Sr-U and limitations of Sr/Ca proxies, while Hathorne is primarily a technical paper. These may not be the most appropriate citations for this context.

Line 285: The authors agree with this statement and changed the references to more classical ones to support this statement.

Line 275 – There seems to be some overlap with content in line 256.

Line 287: The duplicated statement "which was measured at the start of each sequence" has been removed.

Line 282 – As noted earlier, the term "trace elements" may be interpreted differently by the coral paleoclimate community. Clarification may be useful here.

Line 292: For clarity the term "trace" has been removed.

Line 343 – The sentence suggests that the impacts of real-world events could be more severe, but it would be helpful to explain why the metal release levels differ between the experiment and natural scenarios.

The authors agree that the line "However, the input from La Soufrière ash is not representative of average ash deposition." is awkwardly phrased. The intention behind this statement is explained in the following sentence, where the leaching ability of the La Soufrière ash is compared to the global median leaching values presented in Ayris and Delmelle (2012) (lines 354-357). The sentence was changed for clarity (line 354).
As mentioned in **Part I,** the goal of this study was not to replicate the La Soufrière eruption from 2021, but rather focus on a mechanistic and holistic understanding of ash-derived metal cycling within the coral.

Line 368 – The sentence is awkwardly worded, with an unclear subject and inappropriate comma usage. Please revise for clarity.

Changed.

Line 460 – The mechanisms of metal incorporation into the skeleton may be more critical than their seawater concentrations. For example, Mg is more abundant in seawater than Sr, yet is depleted in coral skeletons, whereas Sr is enriched, largely due to selective incorporation. The manuscript does not sufficiently consider such mechanisms for the trace elements discussed.

Please refer to the answer in **Part I.**

Line 509 – This statement largely repeats earlier content, and again the supporting references do not appear to be the most appropriate.

Lines 531-532: References were changed accordingly.

**Anonymous Referee #2:**

The manuscript by Förster et al. details the results of a geochemical study into *Stylophora pistillata* cultured in tanks exposed to volcanic ash. The topic covered in this paper is important because it has implications for the health of corals and for their potential role as archives of past volcanic eruptions. The scope of the study is novel, in particular because it investigates the impact on the geochemistry of three different coral compartments from the same individuals - skeleton, tissue, and symbionts. Together this means the manuscript falls within the scope of *Biogeosciences*. The methods used are appropriate and, for the most part, adequately documented. The text and the figures are clear and concise, whilst the referencing is thorough. I enjoyed reading this manuscript, and I recommend its acceptance following some minor revisions listed below.

The authors thank the referee for their feedback!

**Minor Points:**

- A little more emphasis on the big picture, and why this study is important, should be added to the abstract, and to a lesser extent the conclusion, to stress the significance of the study. I.e. ramifications for coral health, and presumably as an archive of past volcanic eruptions (in relation to the skeleton at least). This is currently lacking.

  The authors agree with the raised point and added more significance in the last sentence of the abstract (lines 29-31) and in the conclusions (lines 645-649).

  Out of interest, do you envisage a scenario where the trace metal concentrations of coral skeletons could be used to estimate the volume of ash dropped on a reef, or would there be too many unknowns, such as oceanographic dynamics?

  As the referee mentioned, real-life reef dynamics are tough to constrain. However, there is ongoing research which compliments the laboratory work presented in this study in the field. James Vincent (also a PhD at the University of Geneva) is analysing coral cores surrounding the island of St. Vincent to assess the applicability of different trace elements in coral skeletons as archives of volcanic eruptions. We have added a sentence at the end of the conclusions hinting towards this further work (lines 647-649).

- Line 73, sentence starting with "Stewart et al. (2020) argues that…": please rephrase, as currently unclear.

  Lines 77-80: The sentence has been rewritten for clarity.

- Line 132, referring to accuracy of solution measurements. You state that recoveries ranged from 87% to 110%. So preconcentration procedures fractionate your solutions? Was this

then corrected for when analysing your samples? If so, please detail this step. If not, then why not?

The authors acknowledge that recovery values deviating from 100% may suggest potential losses or matrix effects. The recoveries for Fe and V were highlighted as they define the lowest and highest recoveries, respectively, based on analyses of the RM CASS-6 and SAFe D1_479. However, given that most of the elements show deviations < 5% from the certified consensus values (as mentioned in line 139) and that our recoveries fall within the range reported by Rapp et al. (2017), we consider our seawater trace metal measurements to be accurate.

- In line 140 you state the culture conditions, then in line 416 you state that the impact of the ash dosing on light and pH was "negligible". Can you expand on this, and state to what degree both of these parameters changed during the ash dosing, and or how long?

  Lines 427-428: The sentence was expanded accordingly to quantify the "negligible" changes after ash addition.

- Line 146, and the clause "natural containing metals with ash derived metals.": please rephrase as this isn't clear.

  Optimized for clarity.

- Please comment on how your ash dosing (2.5 g, three times a week, into your 30 L culturing tanks, line 150) compares to natural analogues. What made you choose this mass and rate, and how common is this dose rate/amount is likely to be in nature? Would this constitute an extreme case of ash-fall, or more a more minor one'?

  Please refer to the answer to Anonymous Referee #1 in **Part I.**

- Section 2.2.2: Please give details on how the major and trace element measurements of the tissue and symbionts were standardized, and the accuracy/precision of these measurements.

  Lines 227-231: A small paragraph containing standardization and accuracy/precision of the "organics" measurements has been added.

- Line 206: deleted the comma after "in this section" as it is unnecessary.

  Changed.

- Line 221: OES and ICP-MS are techniques, not instruments, so please rephase to: "…measured by ICP-OES…and QQQ-ICP-MS…".

  Changed.

- Line 230: were any polishing agents used when grinding the skeleton samples for LA-ICP-MS?

  To avoid introducing potential "metal contaminants", the corresponding author only used pristine Silicon Carbide grinding papers with two different mesh sizes, while a continuous flow of water rinsed and cooled the grinding disc.

- Line 257: please state what values were used for NIST612 when standardizing your data.

  Lines 266: The GeoReM preferred values (Jochum et al., 2011) were used as reference concentrations. Added accordingly.

- Line 258: please state what Ca concentrations were used for internal normalization.

  For internal normalization 40.04 wt.% of Ca was assumed (as for pure calcium carbonate). This information is given in line 281.

- Line 374, sentence starting with "Here, we present…". Please rephrase as currently unclear. I think you may just need to say "…changes in the…"

  This line refers to previously published data on metal concentrations in coral tissue and symbionts (Förster et al., 2024, *STOTEN*). The corresponding author felt it was necessary to clarify the repeated presentation of these findings and did so by noting that the data are presented here in a different context (holobiont metal cycling).

- Line 411/Results section: you quite rightly focus on the geochemical differences between the skeleton/tissue/symbionts cultured in the two treatments, but to put these in context we are missing an assessment of the tank effect within treatment: are there any statistically significant differences between the three compartments grown in the replicate tanks? Similarly, were there any differences between nubbins grown within the same tank?

  Following the referee's suggestion, the corresponding author added a few sentences in section 3.2.2 (lines 433-434 & 446-449) plus a supplementary table (a new Tab. S7a&b) listing the mean metal concentrations of the coral compartments of the various experimental tanks (control tank A and B, and ash-exposure tanks C and D).

  Nubbins were randomly sampled from different mother colonies to limit inter-tank variability. Consequently, this resulted in within-tank variation of geochemical values for the three coral compartments. This within-tank variation (as seen in the newly added tables S7a&b) is systematically larger than the between tank variation in mean values, suggesting the absence of a systematic tank effect.

- Line 548: Chalk et al. 2021 (https://doi.org/10.1038/s41598-020-78778-1) also show this for *S. siderea.*

  Line 570: Reference has been added.

- Line 557: you say that each laser spot reflected a mixture of COC and fibrous aragonite. Is this assumed, or could you actually see the COCs to make certain of this? Please make this clear either way. It seems unlikely that you can be absolutely certain of equal proportions, so could some of your variation be caused by different COC:fibre ratios?

This statement was assumed as the COCs were not visible in the embedded coral apices. However, Von Euw et al. (2017) could show the distribution of COCs in the skeletal features (theca, septa, columella) of *S. pistillata* using polarized light microscopy and revealed COCs with a diameter of a few microns. With a laser spot size of 60 microns, we most likely obtained averaged chemical signals from COCs and fibrous aragonite with no clear COC:fibre ratio. To clarify the unknown proportions of these phases within each spot, an explanatory sentence was added (line 581).

It would be interesting to see if the ash-derived trace metals are preferentially incorporated into one or the other of these structural components.

The corresponding author shares the interest in the chemical and microstructural effect of ash-exposure, and hopes further work can investigate this beyond the imminent end of his PhD.

---

## Author Response (AR2)

**Response to the Reviewers (Round 2)**

Disclaimer: The referenced lines always refer to the revised manuscript WITH tracked changes.

**Associate Editor Chiara Borrelli:**

The authors incorporated the suggestions provided to them during a first round of reviews in a satisfactory manner. The manuscript will be suitable for publication after the authors will address the latest minor revisions they received after this second round of evaluation of their work.

**Anonymous Referee #2:**

The Authors have addressed all my (Reviewer 2) comments adequately, and the significance and importance of the study stands. Following the authors revisions, I have two further points for them to address prior to acceptance of this manuscript.

1) Line 438: please define "PAR".

Line 429: Changed. Since the abbreviation "PAR" only appeared once in the manuscript, we have replaced it with the full term "photosynthetically active radiation" for clarity.

2) Thank you for adding in details regarding the tank effects. In it, you show that there are significant variations in Fe, Cu, and Pb between the control tanks, and in Mn, Fe, Ni, Cu and Zn in the ash-exposed tanks. Bearing in mind a key conclusion was that "Ash exposure enriched skeletal concentrations of V, Mn, Fe, Ni, and Zn while depleting Cu and Pb", the last thing the manuscript needs is clarity on how you concluded this when same treatment tank effects show significant variations in the majority of these elements. Presumably statistical tests demonstrate that variation relating to the treatments is greater than that relating to within-treatment tank effects.

The authors thank reviewer #2 for this comment related to tank-related effects in the chemical composition of coral skeletons and to ensure the clarity of our conclusions. To address this we have: (i) conducted a nested ANOVA test to checked the importance of treatment versus tank; and (ii) performed a Tukey's post-hoc test to detect statistical differences in the skeletal compositions between nubbins of individual tanks. As both tests have not been performed before in the scope of the manuscript, we have added a sentence in chapter "2.4 Statistical Analysis" (lines 319-320). The obtained statistical results are now presented in a newly added supplementary table (Tab. S9) and visualised in a newly added supplementary figure (Figure S1).

The nested ANOVA results (Table S9) indicate that only for copper the tank-related effect is stronger than the treatment effect, but given the tank-related effect is dominant in ash-exposure treatments versus controls, we believe the population effect using the ANOVA test are still valid (Table S8). Consequently, the key conclusion "Ash exposure enriched skeletal concentrations of V, Mn, Fe, Ni, and Zn while depleting Cu and Pb" remains valid as it refers to the average chemical composition of the coral skeletons at the population level (i.e., comparing nubbins from both control tanks with those from both ash-exposed tanks). To improve clarity, we have added a small section summarizing this information (lines 453-468), along with Fig. S1 and Tab. S9.